# Motivation for Physical Activity: Validation of the Dutch Version of the Physical Activity and Leisure Motivation Scale (PALMS)

**DOI:** 10.3390/ijerph18105328

**Published:** 2021-05-17

**Authors:** Wim van Lankveld, Fieke Linskens, Niki Stolwijk

**Affiliations:** Musculoskeletal Rehabilitation Research Group, Institute of Health Studies, HAN University of Applied Sciences, 6503 GL Nijmegen, The Netherlands; fieke.linskens@han.nl (F.L.); niki.stolwijk@han.nl (N.S.)

**Keywords:** Physical Activity and Leisure Motivation Scale (PALMS), motivation, physical activity

## Abstract

Understanding motivation for exercise can be helpful in improving levels of physical activity. The Physical Activity and Leisure Motivation Scale (PALMS) measures distinct goal-oriented motivations. In this study selected measurement properties of the Dutch version (PALMS-D) are determined. Forward-backward translation was used for cross-cultural adaptation. Construct validity of the PALMS-D was assessed in five subsamples completing the PALMS-D and the Behavioral Regulation in Exercise Questionnaire (BREQ-3). The study population consisted of five samples recruited from different populations; samples consisted of runners, hockey players, soccer players, participants in medical fitness, and a sedentary group of young adults with low activity. A total of 733 participants completed the questionnaire: 562 athletes and 171 non-athletes. Exploratory for Analysis confirmed the original eight factors. Internal consistency of the subscales was high, except for Others’ expectations. The a priori determined hypotheses related to differences between athletes participating in different sports were confirmed, as well as the hypothesis related to differences between amateur athletes, patients in medical fitness, and non-active participants. It was concluded that the Dutch version of the PALMS is an acceptable questionnaire with which to evaluate the individual motivation of athletes in the Netherlands, and discriminates between different leisure athletes, patients in medical fitness, and non-active youths.

## 1. Introduction

Physical Activity (PA) is important for both physical and mental health [1]. PA has been defined as engaging in light, moderate or vigorous physical activity, and is related to lower mortality as well as primary and secondary prevention of many chronic medical conditions [1,2,3]. The World Health Organization recommends that adults accumulate at least 150 min of moderate-intensity aerobic physical activity throughout the week, or undertake at least 75 min of vigorous-intensity aerobic physical activity throughout the week [4]. In line with these recommendations, many countries have adopted PA and Sedentary Behavior (SB) recommendations [5,6,7,8]. However, most PA and SB policies have low to moderate effectiveness [6]. Insufficient PA is an important health hazard with high costs to society [9]. Insufficient PA has been reported across countries in the general population [3,10]. About 40% of European adults reported insufficient PA [11], with 46% of the inhabitants never exercising or playing sports [4]. The proportion of people with SB is increasing, and sedentary lifestyle is related to overweight and obesity [12,13,14]. Therefore, to prevent unnecessary deterioration of physical health in the population, levels of PA should be improved, particularly among relatively inactive subgroups [11].

Targeting motivation for PA is considered key to enhancing PA, as motivation is a proximal determinant to behavior [15]. In PA motivation research, the Self Determination Theory (SDT) is frequently used [16]. SDT focusses on differences in the ways in which people’s behavior can be regulated and how these differences are experienced. The highest level of motivation, intrinsic motivation, arises from the willingness to understand, practice and master a task [17]. While intrinsic motivation refers to participating in physical activity for fun and pleasure, extrinsic motivation refers to external rewards or demands [18]. Overall, SDT has improved our understanding of exercise behavior, demonstrating the importance of intrinsic or autonomous regulations in fostering PA [19,20,21,22].

In line with the SDT, the Behavioral Regulation in Exercise Questionnaire (BREQ) was developed [23]. The BREQ measures motivation, defining subscales for amotivation, external regulation, introjected regulation, identified regulation, integrated regulation and intrinsic regulation [24]. As such, the BREQ measures the individuals’ motivation and regulation of PA in general. However, individuals differ in the goal they have regarding participation in PA [25], and such goal-oriented motivation is likely to guide the individual’s choice of PA [26]. For instance, in team sports athletes (as in soccer) being part of a team is likely to be a more important goal of PA compared to athletes participating in individual sports (cf. runners). Such individual goal-oriented motivation is core to many behavioral intervention approaches [20,27,28]. These approaches focus on what the patient can and wants to do to improve his/her own health and wellbeing in the future [28]. Understanding an individual’s goal-oriented motivation for specific PA might have important clinical implications for health professionals. For instance, Physical Therapists (PT) are often confronted with patients with chronic conditions that ask for prolonged and regular PA, as well as patients with insufficient PA. In both cases, enhancing PA is a challenge: in chronic conditions compliance to PA advice is low [29], and people with insufficient PA do not enjoy being active [30,31]. The assessment and understanding of the individual’s goal-oriented motivation is therefore particularly important for the PT as it will help target and improve PA in these patient groups.

To assess an individual’s goal-oriented motivation, the PALMS was developed [32]. The scale measures eight types of motivation (Mastery, Enjoyment, Affiliation, Competition/ego, Others’ expectation, Physical condition, Psychological condition, Appearance). Of these eight subscales, Enjoyment and Mastery factors can be considered as intrinsic motivation, while the other six factors describe extrinsic motivation based on SDT [33]. The PALMS has been translated in several languages [33,34,35,36,37]. In general, the PALMS has shown excellent measurement properties, and was able to distinguish motivation between athletes participating in individual versus team sports [25]. However, the PALMS is as yet not available in Dutch. Furthermore, PALMS has been used primarily in active athletes without sports injuries. As a result, it is not clear whether the questionnaire can be used in non-athletic people often coached by FT, for instance patients with chronic conditions participating in medical fitness rehabilitation, or in individuals with low levels of PA.

Therefore, the objective of this study is (a) to determine measurement properties of construct validity and reliability of the PALMS-Dutch version, and (b) to determine the different types of goal-oriented motivation among healthy athletes compared to recovering patients in medical fitness, and respondents low in PA.

## 2. Materials and Methods

### 2.1. Study Design

The study was conducted in two phases. In the first phase, the PALMS questionnaire [32] was translated into Dutch [38]. The second phase consisted of determining selected measurement properties of the PALMS-D using the COSMIN criteria [39].

Phase 1: Cross cultural adaptation of the PALMS questionnaire.

The original author was contacted and permission was asked and given to translate the original PALMS into Dutch (PALMS-D). The PALMS was adapted for the Dutch language using the Beaton method [38]. First, the original items of the PALMS were translated from English to Dutch by two independent translators. Secondly, an expert panel compared and discussed the different translations until consensus was reached. The expert panel included both translators, the researcher responsible for the project (WvL), and three students participating in the project. The third step included a back translation from Dutch into English of the synthesized version by two different independent translators. The translators worked independently from each other and then compared their translations. Differences found in these translations were reported to the same expert panel. At step four, an expert committee including all four translators and two researchers discussed the final version of the back-translation by e-mail. Content validity of the translated version in Dutch was checked in a small sample of participants.

Phase 2: Determining measurement properties of the PALMS-D questionnaire.

The study was conducted in accordance with the Consensus-based Standards for the selection of health status Measurement Instruments (COSMIN) guidelines [39]. Construct validity (apart from cross-cultural adaptation, this includes structural validity and hypothesis testing) and internal consistency as a measure of reliability of the PALMS-D were determined in a number of different samples of amateur athletes and non-sporters.

### 2.2. Participants

The study population consisted of five samples recruited from different populations; three samples of amateur athletes (runners, hockey players, soccer players), and two special interest groups in need of Physical Therapy coaching (participants in medical fitness, and a sedentary group of young adults with low PA). All participants resided in the Netherlands and Dutch was their native language. All participants were recruited using digital newsletters and social media (Facebook and LinkedIn). For the amateur athletes (runners, soccer players, and hockey players) social media outlets provided by their sports associations were used. Medical Fitness patients were recruited by their physical therapist (PT). A total of 29 PT practices in or around Nijmegen, Netherlands, that provided Medical Fitness to their patients were asked to include patients in the study. Finally, social media were used to include young people with self-reported low PA. In this group, patients were asked to participate when they currently engaged in PA for less than 1 h per week.

### 2.3. Procedure

All participants were informed about all aspects of the study by means of an information letter by e-mail. If participants were interested in participating, they could follow the link in the e-mail to a web-based questionnaire tool (ThesisTools, by ThesisTools, Liessel, Belgium). On the first page of the web-based tool, further information about the study was given and active informed consent was asked. When informed consent was given the participant entered the digitalized questionnaire. All questions had to be completed, so there were no missing values. Completion of the online questionnaire took on average 10 min.

The study was conducted in accordance with the Helsinki declaration and approved by the HAN Ethical Board (CAEO 60.0219).

### 2.4. Measurements

All participants in the study reported their gender and age, and completed two questionnaires: the PALMS-D (Table A1) and the BREQ-3-D.

Physical Activity and Leisure Motivation Scale-Dutch (PALMS-D).

The PALMS-D has eight subscales assessing different types of motivation; Mastery, Enjoyment, Affiliation, Competition/ego, Others’ expectation, Physical condition, Psychological condition, and Appearance. A 5-point Likert scale is used in which 1 stands for ‘strongly disagree’ and 5 stands for ‘strongly agree’. Every subscale exists of five items and subscale scores are computed by summing the scores on the items, with subscale scores ranging from 5 to 25. A higher score on a subscale indicates higher levels of that type of motivation.

Behavioral Regulation in Exercise Questionnaire-D (BREQ-3-D).

The BREQ-3-D consists of 24 statements to assess six forms of motivation; amotivation, external regulation, introjected regulation, identified regulation, integrated regulation and intrinsic motivation. Items are scored on a 5-point Likert scale in which 0 stands for ‘not true for me’ and 4 stands for ‘very true for me’. This questionnaire is the Dutch version of the validated BREQ [23]. Subscale scores are computed by summing the item scores. To ease interpretation for this study, the Relative Autonomy Index (RAI) is computed. The RAI is a single score derived from the subscales that gives an index of the degree to which respondents feel self-determined. To compute the RAI the subscale are weighted (−3 for amotivation; −2 for external regulation; −1 for introject regulation; +1 for identified regulation; +2 for integrated regulation; and +3 intrinsic regulation) [40]. Each subscale is multiplied by its weighting, and these weighted scores are summed. A higher score indicates higher levels of self-determination.

### 2.5. Data Analysis

The participants’ responses were entered into a database. Statistical analyses were performed using SPPS version 26.0 (SPPS Inc., Chicago, IL, USA). Gender is described as percentage male/female in each sample. Associations between continuous variables is calculated using Pearson correlations (r). According to Cohen (1988), an absolute value of r as 0.1 is classified as small, an absolute value of 0.3 is classified as medium and of 0.5 is classified as large [41]. Differences in samples will be tested using T-test for independent samples in continuous variables, and χ^2^ in dichotomous variables. Differences between groups on the dependent PALMS-D scales were tested using Multivariate Test of Variance (MANOVA). Significance of difference between groups was tested using F value for Pillai’s Trace. A value lower than 0.05 indicates that the groups differ significantly with respect to the dependent variables. In addition, differences between groups on each of the eight PALMS scales were tested in a univariate analysis.

Structural validity of the PALMS-D version was explored with exploratory factor analysis (EFA). Principal component analysis was used as a dimension reduction technique. Varimax rotation with maximum likelihood extraction was used. Using the Kaiser criterion (Eigenvalue > 1), it is expected that eight factors will be identified similar to the subscales in the original scale.

Reliability was determined based on scale analysis of the combined data from five samples, First, mean sores, standard deviation (SD), range of observed scores of the PALMS-D subscales, as well as skewness were calculated. Skewness is a measure of symmetry of frequency distribution, and values between −1 and +2 indicate normal univariate distribution. Cronbach’s Alpha as a measure of reliability was computed for each PALMS-D subscale as a measure of internal consistency. A Cronbach’s Alpha > 0.75 is considered as good [42].

Hypothesis testing was used to test for further construct validity. To this end a number of a priori formulated hypotheses were formulated (Table 1), based on the assumed relation between PALMS-D subscales and levels of self-determination assessed using the BREQ-3-RAI. It was expected that PALMS-D subscale Enjoyment will have strong association with the BREQ-3-RAI score, and a negative correlation with the subscale Others’ expectation. Another set of hypotheses were based on the assumption that athletes involved in different sports will have different goal-oriented motivations. For instance, it was expected that athletes participating in individual sports (e.g., runners) report different types of goal-oriented motivation compared to athletes participating in team sports (as in soccer). Finally, it was expected that on average the Medical Fitness and Sedentary Behavior groups would differ in goal-oriented motivations from each other and from the active sporters. The construct validity of the scale is considered good when > 75% of the hypothesis can be confirmed [42].

## 3. Results

Phase 1: Cross cultural adaptation.

Two native Dutch people fluent in English translated the items into Dutch and discussed consensus with the main researcher (WvL). Next, two native English speaking persons with Dutch as a second language back translated the consensus Dutch language items into English. The expert panel compared the end result with the original items. No major differences in backward translation were identified. Only some minor changes, not affecting the semantic integrity of items, were considered. This version was tested in 29 respondents (amateur runners; female/male = 62%/38%; average age = 22.14 years). Feedback resulted in further clarification of the introduction of the questionnaire. The original items of the PALMS and their translation in Dutch (PALMS-D) are given in Appendix A.

Phase 2: Measurement properties of construct validity and reliability.

Data from five study samples were combined to analyze a total of 733 participants. Table 2 shows the characteristics of the groups of participants in the study.

As well as differing in number of participants, groups also differed in gender and average age. Gender differences between groups was statistically significant (χ^2^ = 62.8, df = 4, *p* < 0.001). Groups were statistically different in average age (ANOVA F = 228.31, df = 4, *p* < 0.0001). Because samples differed in both gender and average age, these demographic variables will be considered as confounders in hypothesis testing.

### 3.1. Structural Validity

Next, EFA using Varimax rotation was conducted on all 40 items of the PALMS-D. Results are depicted in Table 3, in which the rotated factor structure is given. To ease interpretation, items are grouped into the original PALMS subscales. These groups of items are ranked according to the percentage of variation explained by that factor in the EFA. Within each factor, subscale items for that factor are ranked in order of magnitude (highest factor loading presented first). Only factor loading higher than |>0.40| are depicted. Factor loadings |>0.40| for items on the other factors are depicted in red.

Varimax rotation with maximum likelihood extraction resulted in eight factors: Enjoyment, Mastery, Affiliation, Competition/ego, Others’ expectation, Physical condition, Psychological condition, Appearance. These eight factors accounted for 67.5% of the total variance. For seven of the eight factors, the item loadings were in accordance with expectations, or the scale to which these items were supposed to contribute. The items related to Others’ expectations do not show item loadings as expected. Item loading on factor 8 varied from 0.21 to 0.77, and 3 items had loadings >0.40 on other factors. In line with previous research, we chose to keep the same factor structure as in the original PALMS [25,35].

### 3.2. Internal Consisctency of the PALMS-D Subscales

Table 4 shows summary statistics of the PALMS-D subscales. All scales have a theoretical range from 5–25.

On each of the PALMS-D subscale scores, observed range is equal to the equal range of 5–25, and skewness for each of the subscale is within range of normality. The internal consistency values of the subscales are all high, ranging from 0.80 to 0.92, with the exception of subscale Others’ expectations which has an internal consistency of 0.51.

Based on the average PALMS-D subscale score Physical condition and Enjoyment have the highest scores. In this sample, Competition and Others’ Expectation have the lowest relative scores.

### 3.3. Hypothesis Testing

Groups differ in gender and age, therefore gender and age were considered as confounders prior to hypothesis testing. Gender was related to three of the PALMS-D subscales. Compared to women, male participants on average scored higher on the subscale Competition (average scores 12.07 95%CI 11.50–12.61; versus 9.99 95% CI 9.55–10.37; T = 6.2, *p* < 0.001), and lower on the subscale Appearance (average scores 15.98 95%CI 15.4–16.5; versus 17.73 95%CI 17.3–18.2, T = −5.0, *p* < 0.001). Age was weakly correlated to Competition (r = −0.26, *p* < 0.001, and with Appearance (r = −0.22, *p* < 0.01). Although significant, these associations indicate a weak association between the age of the participants and Competition and Appearance motives. Because samples differed in both gender and average age, these demographic variables were included in the MANOVA analysis.

A priori determined hypotheses about the relation between PALMS-D and BREQ-3-RAI (hypothesis 3–5) were confirmed. In the last column of Table 4, Pearson correlations are given between the different PALMS-D subscales and the BREQ-3-RAI.

Next, differences between groups of participants on the PALMS-D scales were studied. Table 5 shows average scores and 95% Confidence Interval for each group of participants on the eight PALMS scales.

A MANOVA was performed with gender and age as covariates testing the differences between groups on the eight PALMS scales (hypothesis 6). Using Pillai’s trace, there was a significant difference between the five groups in sports motivation on the PALMS subscales (Pillai’s trace = 0.57; F = 15.2, df = 32; *p* < 0.001). Post hoc univariate analysis showed significant differences between groups for each of the eight PALMS scales (all *p* < 0.0001) confirming hypothesis 6.

Table 5 gives 95%CI for each sub-group of participants on each of the PALMS-D subscales enabling comparison between groups. On average, team sporters reported higher scores on the PALMS-D subscale Affiliation compared to the individual sporters (hypothesis 7). As expected, the average scores of the medical fitness group on PALMS-D subscale Competition is lower than the average score for the amateur athletes (hypothesis 8 confirmed). Hypothesis 9 is only partly confirmed: participants in the Medical Fitness group reported higher levels of PALMS-D Physical Condition when compared with the sport athletes group. However, runners reported even higher levels of Physical Condition. Finally, participants in the low PA group reported considerably lower levels of Enjoyment compared to either of the amateur athletes groups (hypothesis 10 confirmed).

## 4. Discussion

Understanding the individual’s motivation for PA can be helpful to a coach, trainer, or a physical therapist in order to stimulate exercise. This study reports measurement properties of a Dutch questionnaire to measure goal oriented motivation for PA. Overall, the measurement properties of the PALMS-D are sufficient to assess levels of PA motivation in different subgroups. These findings confirm earlier findings reporting structural validity and internal consistency of the PALMS subscales [26,33,34,36].

With regard to structural validity there is some discussion in the international literature on the factor structure of the PALMS [35]. In this study, the EFA on the PALMS-D replicated the eight factors structure, comparable to the factors reported by Morris and Rogers [43]. However, the items of the Others’ expectation subscale did not perform well in the factor structure. Similar problems with the Others’ expectation dimension were observed in other studies reporting cross cultural adaptations of the PALMS [33,37]. It seems that the Others’ Expectation subscale consists of items referring to distinct dimensions. Item 18, for example, ‘To manage medical condition’, showed higher factor loading on the factor Physical condition, instead of Others’ expectations. Item 1 and 7, both items on financial support, have poor factor loading on the subscale factor. The lack of stability of the dimension Others’ expectations might also explain its relatively poor internal consistency. In the study of Zarei et al. [33], Others’ expectations also showed a lower internal consistency compared to the other subscales. Although this scale did not perform well, we chose to keep the same subscale structure as in the original PALMS [25,35] as it allows the comparison of results from different versions or countries in future studies. Of the a priori formulated hypotheses, more than 75% were confirmed, indicating good construct validity.

The second aim of this study was to compare different types of goal-oriented motivation among healthy athletes with motivation in patients in medical fitness and respondents low in PA. In the statistical analysis, the five groups of participants differed on each of the PALMS-D subscales, indicating that the PALMS-D is able to measure differences in goal-oriented motivation between groups. Moreover, four a priori formulated hypotheses related to differences in goal-oriented motivation between groups were confirmed. The data in our study confirmed that patients participating in the medical fitness group (hypothesis 8 and 9) were less motivated by competition motivation than by motivations of physical condition. Furthermore, in accordance with previously reported findings, participants selected based on their low PA reported lower levels of Enjoyment. Therefore, it can be concluded that the PALMS-D is not only a valid and reliable instrument, but also has additional value in measuring distinction in goal-oriented motivation between individuals.

This study is not without its limitations. First, all participants were recruited using open calls to participate placed in digital newsletters and social media. This means that a selection bias cannot be ruled out: it is likely that some individuals are more prone to respond to social media than others. This is no problem in determining selected measurement properties of The PALMS-D in this study. However, the reported conclusions based on the differences between groups reported should be interpreted with care, as it is not clear whether participants are representative of the entire population in each group. Another limitation is that questionnaires were self-reported which makes reporter’s bias possible, invoking socially desirable answers and thereby affecting the conclusion on the type of motivation for PA per group. This might be the case in particular for the participants low in PA, as PA is highly promoted and desirable in western societies.

Using the PALMS to assess motivation to engage in PA might have some advantages compared to assessment with other instruments. Motivation is a proximal determinant to behavior [15], but the conceptualization and assessment of motivation is complex. For instance, the BREQ originally conceptualized motivation as a continuum of self-determination, ranging from amotivation to intrinsic motivation [16]. However, this one dimensional nature of the BREQ in the measurement of motivation has been disputed [44]. Both on conceptual and statistical grounds, a multidimensional conceptualization of motivation as assessed with the BREQ has been proposed [44]. Because the dimensionality of the BREQ has as yet not been resolved, the meaning of the individual’s score on the BREQ scales is hard to interpret. Others have taken a different approach in the conceptualization of motivation, putting goal orientation as central in the assessment of motivation. For instance, the Exercise Motivation Inventory [45] measures 14 reasons for a person to exercise. In a similar way, the PALMS is designed to measure clearly distinct goal-oriented motivations for PA in general. In line with previous studies, our study showed that individual differences exist between preferences to engage in a particular physical activity modus [46]. Moreover, our study showed that individuals might have distinct goal-oriented motivations simultaneously. The scoring of the soccer players is a good example: on average they have high scores on Mastery, Enjoyment, Affiliation, Competition and Others’ expectation simultaneously. This ability of the PALMS to assess distinct goal-oriented motivations simultaneously might be of help in clinical practice. Sports frequency is higher when participants engage in settings that better fit their motivation and goals [47]. Therefore, the PALMS can help therapists or doctors to provide insight into the motivation of patients when coaching them towards a change in behavior. In behavioral intervention approaches, for instance Motivational Interviewing [27], it is important to discuss the individuals motivation to change, in order to help the patient to engage in activities that best suit the specific motivational goals.

## 5. Conclusions

The PALMS-Dutch Version is an acceptable questionnaire to evaluate the individual motivation of runners, hockey players, football players, medical fitness groups and low-activity in the Netherlands. It can be an important tool to describe differences in motivation and can help to stimulate people to enhance their PA. In research, the PALMS-D Version can be used to compare different types of motivation and compare different subgroups. However, the subscale Others expectation does not perform well and should be interpreted with care.

Key Findings:➢The PALMS-D is a valid instrument for assessing individual’s motivation to be active.➢The PALMS-D discriminates in motivation between leisure athletes, patients in medical fitness, and non-active youths.➢The PALMS-D sub-scale “Others’ Expectations” has poor measurement properties and should be interpreted with care.➢The PALMS-D measures “Appearance” and “Competition” as motivations not addressed in the BREQ-3-D.

## Figures and Tables

**Table 1 ijerph-18-05328-t001:** A priori formulated hypothesis for construct testing (confirmation/rejection = +/−).

1	The PALMS-D is expected to have eight dimensions.	+
2	The PALMS-D eight subscales will have medium intercorrelations (0.30 < r < 0.50.)	+
3	The PALMS-D subscale Enjoyment will show a large correlation (r > 0.50) with BREQ-3 Relative Autonomy Index.	+
4	The PALMS-D subscale Mastery will show moderate positive correlation (r > 0.30) with BREQ-3 Relative Autonomy Index.	+
5	The PALMS-D subscale Others’ expectation will show a reversed medium correlation (r > 0.30) with BREQ-3 Relative Autonomy Index.	+
6	Different groups of participants differ in goal-oriented motivation assessed with the PALMS sub-scores.	+
7	Average scores of amateur sporters involved in team sports (hockey players and football players) will score higher on subscale Affiliation of the PALMS compared to runners.	+
8	Average score for Competition in the amateur sporters will be higher compared to participants in the Medical fitness group.	+
9	Average score in the Medical fitness group on Physical condition will be higher compared to amateur athletes.	+/−
10	Average score for low activity participants on Enjoyment will be lower compared to amateur athletes.	+

**Table 2 ijerph-18-05328-t002:** Demographic characteristics of the study population (n = 733).

Subsample	n	Gender (% Male)	Age (Mean ± SD)	Age (min–max)
Active Athletes				
Runners	447	42.5	42.59 ± 13.72	17–72
Hockey players	81	48.1	24.97 ± 5.84	18–45
Football players	34	100	24.42 ± 6.35	19–49
Special Interest Groups				
Medical fitness	91	31.9	69.30 ± 11.19	34–86
Low-activity	80	23.8	24.66 ± 4.83	18–35
Total	733	42.4	41.2	17–86

n = number of participants; SD = Standard Deviation; min–max = minimum and maximum age.

**Table 3 ijerph-18-05328-t003:** Summary of Exploratory Factor Analysis of the PALMS-D using varimax rotation.

Label	Item	Factor Loading							
		1	2	3	4	5	6	7	8
Enjoyment	37	0.79							
	25	0/78							
	34	0.78							
	13	0.65							
	03	0.45							
Affiliation	38		0.87						
	20		0.85						
	08		0.84						
	30		0.84						
	04		0.79						
Appearance	23			0.85					
	32			0.84					
	11			0.76					
	36			0.75					
	40			0.72					
Competition/ego	29				0.85				
	17				0.82				
	06				0.78				
	27				0.74				
	39				0.73				
Physical condition	15					0.81			
	10					0.80			
	12					0.76			
	28					0.72			
	33					0.50			
Psychological condition	22						0.84		
	09						0.83		
	14						0.76		
	35						0.71		
	02	0.54					0.54		
Mastery	16							0.80	
	24							0.68	
	19							0.61	
	05							0.61	
	31							0.56	
Others’ expectations	01								0.77
	07								0.65
	26	−0.52							0.44
	21	−0.54							0.38
	18					0.51			0.21
% explained variance		25.00	12.18	9.15	5.78	5.19	4.41	3.15	2.68
Eigenvalue		10.00	4.87	3.66	2.31	2.08	1.77	1.26	1.07

**Table 4 ijerph-18-05328-t004:** Summary statistics of the PALMS-D subscales (N = 733); and correlation with the BREQ-RAI.

PALMS-D Subscales	Observed Range	Skewness	Average Score (SD)	Internal Consistency (α)	Correlation with BREQ-3-RAI
Mastery	5–25	−0.58	17.4 (3.7)	0.79	0.35 **
Enjoyment	5–25	−1.20	19.8 (3.9)	0.88	0.69 **
Psychological condition	5–25	−0.60	18.1 (4.4)	0.87	0.38 **
Physical condition	5–25	−1.86	21.5 (3.3)	0.89	0.49 **
Appearance	5–25	−0.51	16.9 (4.8)	0.89	0.17 **
Others’ expectation	5–25	−0.7	10.0 (3.2)	0.51	−0.31 **
Affiliation	5–25	−0.48	16.1 (5.1)	0.92	0.20 **
Competition	5–25	0.64	10.8 (4.6)	0.87	−0.02

SD = Standard Deviation; α = Cronbach’s Alpha; ** = *p* < 0.05.

**Table 5 ijerph-18-05328-t005:** Average scores (95% CI) for each group of participants on the eight PALMS-D scales.

Scale	Runners (N = 447)	Hockey (n = 81)	Soccer (n = 34)	Fitness (n = 91)	Low Activity (n = 80)
Mastery	17.4 (17.1–17.8)	18.0 (17.4–18.7)	18.4 (17.6–19.3)	17.9 (17.1–18.7)	15.7 (14.8–16.5)
Enjoyment	20.6 (20.2–20.9)	20.9 (20.5–21.4)	20.7 (19.8–21.6)	17.1 (16.2–18.0)	16.4 (15.3–17.4)
Affiliation	15.6 (15.1–16.1)	19.8 (19.3–20.4)	19.5 (18.2–20.8)	15.6 (14.6–16.5)	14.8 (13.7–15.9)
Competition	10.7 (10.06–10.89)	13.8 (12.8–14.8)	15.4 (14.2–16.6)	8.9 (8.1–9.6)	10.4 (9.4–11.4)
Others’ expectation	9.3 (9.0–9.6)	9.8 (9.2–10.5)	12.4 (11.2–13.6)	11.7 (11.0–12.4)	11.1 (10.3–11.9)
Physical condition	22.3 (22.0–22.6)	20.4 (19.8–20.9)	19.3 (18.3–20.3)	21.5 (20.9–22.2)	19.3 (18.5–20.2)
Psychological condition	19.1 (18.7–19.5)	17.7 (17.0–18.5)	17.7 (16.7–18.7)	14.8 (13.9–15.7)	16.9 (15.8–17.9)
Appearance	17.9 (17.4–18.3)	16.9 (15.9–17.9)	16.8 (15.5–18.9)	13.4 (12.4–14.4)	16.4 (15.3–17.5)

## Data Availability

Data of this study are available by reasonable request form the authors.

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
