# Peer review of "Motivation for Physical Activity: Validation of the Dutch Version of the Physical Activity and Leisure Motivation Scale (PALMS)"

_ijerph, 2021, doi:10.3390/ijerph18105328_

Round 1

Reviewer 1 Report

The manuscript entitled “Motivation for physical activity: validation of the Dutch version of the Physical Activity and Leisure Motivation Scale 3 (PALMS)” is founded on an important question.  In this study, the authors sought to determine if the PALMS-Dutch version of the scale is a valid and reliable measure.  The PALMS-D appears to be a valid and reliable scale.  However there are some questions for the authors.

Page 8, Line 251

Identify what the symbols (**) indicate in Table 4.  The table cannot stand on its own as presented.

Page 11, Line 379

If the “Others expectation” sub category of the PALMS-D does not correlate with the BREQ-3-RAI, how can the authors justify leaving it in?

Page 11, Line 380

The subcategory of “appearance” also failed to meet the medium or moderate correlation threshold of r > 0.30, yet it was not mentioned in the conclusion.

Author Response

Reviewer 1:

Page 8, Line 251

Identify what the symbols (**) indicate in Table 4.  The table cannot stand on its own as presented.

As suggested by the reviewer, we have identified the meaning of the symbol in Table 4. (Table 4, line 251)

Page 11, Line 379

If the “Others expectation” sub category of the PALMS-D does not correlate with the BREQ-3-RAI, how can the authors justify leaving it in?

The reviewer is highlighting a point of concern: the scale “Others Expectations” does not perform well in the Factor Analysis, and has a reversed correlation with BREQ-3-RAI.

The PALMS is an international accepted instrument to assess individual’s motives to be active. Our aim was to report aspects of validation on the Dutch version of the PALMS. In handling the problems with structural validity, we took the same approach reported in earlier translations.  Therefore we left the original factor structure and sub scaling intact (see line 252). The scale “Others Expectation” has a reversed correlation of e = -31 with BREQ-3-RAI. We did not formulate any hypothesis with regards to this relation, therefore there was no reason to excluded this scale based on the reversed correlation with the BREQ-3-RAI. 

However, to justify this choice in the text we added the following sentence.

Line 253: The choice to keep the original structure of the PALMS intact enables researchers to compare these findings with their own results using the PALMS in other languages.

To warn the reader for this shortcoming we have  added this topic in the Key findings.

Line…:

  • The PALMS-D sub-scale “Others Expectations” has poor measurement properties and should be interpreted with care.

Page 11, Line 380

The subcategory of “appearance” also failed to meet the medium or moderate correlation threshold of r > 0.30, yet it was not mentioned in the conclusion.

The reviewer is right in observing that appearance  is not mentioned in the conclusion as this correlation is no part of the hypothesis to be tested.  However, in the revised manuscript we have pointed out this topic by formulating one key finding with regards to the difference between motivation assessed with the PALMS-D and motivation assessed with the BREQ-3-D. Now line 404:

The PALMS-D measures “Appearance” and “Competition” as motivation not addressed in the BREQ-3-D.

Reviewer 2 Report

Dear Authors,

I have reviewed the following manuscript “Motivation for physical activity: validation of the Dutch version of the Physical Activity and Leisure Motivation Scale (PALMS)” I have following comments:

Comments#

PALMS is an old validated assessment to evaluate individual’s goal-oriented motivation for physical activity. Previously, PALMS has been translated in multiple languages and largely used utilized in assessing the physical activity motivation in the active athletes without sports injuries.

In this study, researchers tried to demonstrate whether the PALMS questionnaire can assess the physical activity motivation in non-athletes, often coached by Fitness trainers. These individuals include patients with chronic conditions undergoing physical rehabilitation or individual with low levels of physical activity in general.

Two important questions were attempted to determine the reliability of the PALMS-Dutch version in evaluating the various types of goal-oriented motivation among healthy athletes compared to recover recovering patients and respondents low in physical activity.

In th current study, researchers found PALMS an important tool to describe differences in motivation and can help to stimulate people to enhance their physical activity. The Dutch version of the PALMS can be used to compare different types of motivation and compare different subgroups.

To increase the readability of this reports, I suggest authors to write their keys findings very briefly in the conclusion section.

English grammar, spelling check and proofreading is required.

After the implementation of the minor comments, I suggest this study to be considered for the publication in the IJERPH.

Author Response

Reviewer 2:

To increase the readability of this reports, I suggest authors to write their keys findings very briefly in the conclusion section.

As suggested by the reviewer we have formulated some key findings at the end of the conclusion.

Line 398-405

Key Findings:

  • The PALMS-D is a valid instrument to assess individual’s motivation to be active.
  • The PALMS-D discriminates in motivation between leisure athletes, patients in medical fitness, and non-active youths.
  • The PALMS-D sub-scale “Others Expectations” has poor measurement properties and should be interpreted with care.
  • The PALMS-D measures “Appearance” and “Competition” as motivation not addressed in the BREQ-3-D.

English grammar, spelling check and proofreading is required.

Reviewer 3 Report

Thanks for submitting your manuscript. It generally reads well but it can be improved further if you consider the following recommendations-

Page 2 line 90-91, mention the study time and duration

Page 2 line 95, provide the reference of the original publication

Page 3 line 100, what do you mean by back translation?

Page 3 line 103, who were the members of the expert committee?

Page 3 line 116, What is PT?

Page 3 line 122, Physical Therapy is used here. Is it the same as PT?

Mention the study place/setting.

Page 3 line 161-163, delete ' with a level of significance ......95% Confidence interval'.

Author Response

Reviewer 3:

Page 2 line 90-91, mention the study time and duration.

We did indeed not mention the time  investment of participants in the study and we are thankful to the reviewer for pointing this out.  However, we felt that this information is best given in the methods section. Therefore we included the sentence Line 140:

Completion of the online questionnaire took on average 10 minutes.

 Page 2 line 95, provide the reference of the original publication

As suggested we have included the reference.

Page 3 line 100, what do you mean by back translation?

To clarify we have now included the words: Now in line 104:

from Dutch into English

Page 3 line 103, who were the members of the expert committee?

To clarify the word Same in included in line 107

Page 3 line 116, What is PT?

To clarify we have no replaced PT by Physical Therapy (now line 120)

Page 3 line 122, Physical Therapy is used here. Is it the same as PT? Mention the study place/setting.

Line 126 now reads:

A total of 29 PT practices in or around Nijmegen, the Netherlands that

Page 3 line 161-163, delete ' with a level of significance ......95% Confidence interval'.

As suggested we deleted these lines.